# Accessing Fungal Contributions to the Birch Effect: Real-Time Respiration from Pore-Scale Microfluidics

**DOI:** 10.3390/microorganisms12112295

**Published:** 2024-11-12

**Authors:** Yi-Syuan Guo, Karl K. Weitz, Aramy Truong, Adam G. Ryan, Leslie M. Shor, Arunima Bhattacharjee, Mary S. Lipton

**Affiliations:** 1Pacific Northwest National Laboratory, Richland, WA 99354, USA; yi-syuan.guo@pnnl.gov (Y.-S.G.); karl.weitz@pnnl.gov (K.K.W.); aramy.truong@pnnl.gov (A.T.); adam.ryan@pnnl.gov (A.G.R.); 2Department of Chemical & Biomolecular Engineering, University of Connecticut, Storrs, CT 06269, USA; leslie.shor@uconn.edu

**Keywords:** Birch effect, soil fungi, drought, real-time mass spectrometry, microfluidics

## Abstract

Drying and rewetting of soil stimulates soil carbon emission. The Birch effect, driven by these cycles, leads to CO_2_ efflux, which can be monitored using real-time mass spectrometry (RTMS). Although soil fungi retain water during droughts, their contribution to CO_2_ release during drying–rewetting cycles remains unclear. In this study, we present the first demonstration of integrating micromodels with RTMS to monitor the Birch effect by simulating drought and rewetting. Micromodels were inoculated with axenic fungal culture and dried to assess moisture retention. After drying, RTMS quantified CO_2_ release upon rewetting with H_2_^18^O mixtures. Our results showed that soil fungi released CO_2_ upon rehydration and immediately utilized the external water source at the pore scale by generating subsequent ^46^CO_2_. This work is the first to integrate RTMS with microsystems to investigate pore-scale biogeochemistry and the involvement of fungi in the Birch effect.

## 1. Introduction

Drying and rewetting of soils is a common phenomenon in dryland ecosystems, where prolonged periods of drought are succeeded by precipitation events. Drylands also store 32% of terrestrial soil organic carbon and the occurrence of drying–rewetting cycles causes the release of the stored carbon in soils in the rapid evolution of CO_2_, a phenomenon well known as the Birch effect [1]. The magnitude of the CO_2_ efflux is likely related to the amount of stored carbon, microbial biomass as well as soil pore characteristics. Moreover, the mechanisms that regulate the Birch effect in soils will likely change as a result of climate change [2,3].

Several specific mechanisms leading to the Birch effect remain unclear [4]. Hypotheses regarding CO_2_ efflux from soils after drying–rewetting events can be categorized into biotic and abiotic factors. Abiotic factors focus on CO_2_ trapped in soil pores, the solubilization of carbonates, or CO_2_ released through chemical reactions with minerals [2]. On the other hand, biotic factors emphasize microbial metabolism of compounds that are either readily accessible or not easily available to microbes in the local environment. To investigate these factors influencing CO_2_ dynamics, researchers have examined CO_2_ efflux using real soil samples to analyze key contributions from the soil–microbe system. Singh et al. pointed out that differences in soil texture influence CO_2_ release by altering bacterial abundance and microbial access to substrates [3]. Recent studies have also reported the accumulation of osmolytes in microbial communities during dry periods [5,6]. While some microbes accumulate compatible solutes during drying, others may die due to drying, rewetting, or a combination of both conditions [2,7]. During drying and rewetting cycles, biotic and abiotic factors are interdependently involved in the CO_2_ release. Abiotic factors, such as chemical deposition and the breakdown of soil aggregates, result in CO_2_ release by modifying the substrate availability to microbes. Meanwhile, at pores, microbes recover from dormancy due to rewetting, activate metabolic processes, and contribute to CO_2_ respiration, likely modulating pore-scale biogeochemistry [3,8]. Such a dynamic interplay of abiotic and biotic factors synergistically contributes to CO_2_ efflux in the ecosystem during drying and rewetting cycles.

To uncover these daunting interactions from the soil pore–microbe interface, we developed an RTMS manifold that provides high-resolution, real-time data for multiple gases to understand better gas exchanges in biological and environmental systems [9]. The setup consists of a main chamber with a gas inlet and a gas outlet connected to the MS instrument, used to monitor bacterial metabolism, plant growth, and soil respiration. The outflowing gas is measured with high precision, multiple times per second, and could be tracked over hours or even days. When applied to soil respiration studies, RTMS captures the microbiome’s immediate response to rewetting in soils from various locations. Here, the CO_2_ release trend was directly linked to the soil’s natural properties, including both biotic and abiotic factors. To further decouple the microbial metabolism involved in CO_2_ release, ^13^C-labeled glucose was added as a carbon source to soil samples to determine if the rate and sequence of metabolism depended on nutrient accessibility upon rewetting [10]. The results of this follow-up study indicated that the immediate respiration of ^12^CO_2_ originated from native carbon sources, likely intracellular or present in the direct environment of the soil samples. The later respiration of ^13^CO_2_ confirmed the utilization of the external carbon source amendment. However, soil moisture levels were maintained within the soil pore structure as capillary water, potentially by the physical structures of soil aggregates [11] or the production of microbial exopolysaccharides (EPS) and plant secretions [12]. These can serve as additional carbon sources for microbes within the soil pores. Thus, it became necessary to disentangle these intricate factors underground. A simplified approach that reduces the complexity of soils while providing real-time feedback on microbial activity is required to investigate the fundamental mechanisms related to microbial contributions to the Birch effect.

In this study, we applied microfluidics that mimic soil porosities and aggregate size distributions to provide direct observation of microbial growth in an axenic fungal system, which has not yet been investigated in the context of CO_2_ release upon rewetting. Our primary goal is to expand the application of real-time mass spectrometry (RTMS) to microfluidics platforms designed to replicate simplified soil structures, enabling the quantification of microbial contributions to the Birch effect at the pore scale. Here, we demonstrate a well-controlled technique for monitoring gas flow at the soil pore scale in real time. Specifically, we investigated whether soil fungi release CO_2_ upon rewetting, serving as an indication of microbial activation to the soil–CO_2_ cycles. *Fusarium* sp. strain DS682 [13], a well-characterized fungal isolate that has been reported to facilitate moisture and mineral-derived nutrient transport [14], was inoculated into soil-mimicking microfluidics, subjected to varying drought periods, and then rehydrated using H_2_^18^O to assess the fungi’s ability to utilize the introduced water source (Figure 1A). Additionally, we fabricated a custom RTMS assembly designed to accommodate multi-purpose microfluidics platforms, enabling the real-time monitoring of microbial activities at the pore scale (Figure 1B,C).

## 2. Materials and Methods

### 2.1. Fungal Strain and Growth Media

The soil fungus *Fusarium* sp. strain DS682 was used for this study [13]. Briefly, this fungal isolate was cultured on MSgg (minimal salts with glycerol and glutamate) agar plates [5 mM potassium phosphate (pH 7), 100 mM morpholinepropanesulfonic acid (MOPS; pH 7), 50 µM FeCl_3_, 2 mM MgCl_2_, 50 µM MnCl_2_, 1 µM ZnCl_2_, 2 µM thiamine, 700 µM CaCl_2_, 0.5% glycerol, 0.5% glutamate with 2% agar] by inoculating 100 µL of spores from frozen stocks. The inoculated MSgg plates were incubated at 25 °C for 7 days before removing fungal biomass for microfluidics use.

### 2.2. Microfluidics Design and Microfabrication

Soil-mimicking microfluidics, emulated soil micromodels (ESMs), were used for this study. Briefly, emulated soil micromodels consist of three parallel channels featuring a 1 mm × 10 mm × 35 μm (width, length, height) microstructured region. Each microstructured region represented a pseudo-2D geometry that mimics a 2D slice of the solid phase of a simulated sandy loam soil. The key features of the micromodels illustrate the realistic pore-scale soil geometry and the ease of direct observation for the progress of the air–water interface over time. Additional details on the validation and use of ESMs can be found in previous publications [15,16,17,18].

Micromodels were fabricated using conventional photolithography and soft lithography methods. Clean, 4-inch diameter silicon wafers were spin-coated with SU-8 2025 photoresist (MicroChem, Westborough, MA, USA) at 2250 rpm, soft-baked on a hotplate, exposed to a 365 nm light at a dose of 150 mJ per cm^2^, post-exposure baked on a hotplate following the user manual’s protocol, developed and rinsed by isopropyl alcohol (VWR, Radnor, PA, USA) and blown by a nitrogen gun. After being given a hardbake at 180 °C for 5 min, the fabricated wafers reached a final height of 35 ± 1.0 µm. Fabricated masters were then treated with air plasma (2 min) prior to silanization [trichloro (1H,1H,2H,2H-perfluoro-n-octyl) silane, 90 °C, 120 min] to prevent polydimethylsiloxane (PDMS, Sylgard 184, Dow Corning, Midland, MI, USA) adhesion to the microfluidics master during the PDMS molding process. The PDMS base and curing agent were fully mixed in a 10:1 ratio and cast over the silanized master, degassed in a vacuum chamber, and then cured at 70 °C for at least 2 h before each inoculation.

### 2.3. Microfluidics Use and Fungal Incubation

PDMS devices in pairs were cut out and trimmed using a scalpel and punched using a 4 mm biopsy punch (Integra^®^ Miltex, Mansfield, MA, USA). PDMS castings were cleaned with tape, and glass slides were cleaned with isopropyl alcohol. Once clean, the PDMS devices were air-plasma-treated (1 min), bonded onto plasma-cleaned glass slides, and placed into the oven to secure device bonding (70 °C, 5 min). After curing and sealing, the PDMS microdevices on glass slides were degassed for 15 min. Once weighed and scanned using microscopy, each device was loaded with 50 µL of MSgg media to both ports using pipette tips. The devices with media were ready to use after UV sterilization for 30 min. Later, a 1.5 mm fungal agar plug was created using a 1.5 mm biopsy punch with a plunger (Integra^®^ Miltex, MA, USA). The fungal plugs were placed with the fungal side facing down in both ports of each sterile device with the 1 cm^2^ PDMS as cover to reduce evaporation. To minimize the solution evaporation during the 10-day incubation, damp tissues were added to the devices or replenished periodically. The Petri dishes with the devices were covered with a lid and sealed with a 2-layer parafilm to prevent contamination and dehydration during incubation. All the treatments were conducted in triplicates. Fungal growth in the channels was monitored to confirm its hydration status using a Nikon inverted Ti-U microscope (Nikon, Melville, NY, USA) with a 10× objective under an EpiLight source. A detailed scheme of microfluidics applications and an example image are illustrated in Figure 2.

### 2.4. Micromodel Drying Experiments, Image Acquisition, and Image Analysis

After a 10-day fungal incubation, the soil micromodels were dried to confirm decreasing saturation, simulating a drought environment. Air infiltration in each channel was monitored over time through the microstructured regions using an epi light as an illumination source and a 10× objective. Overlapping frames covering the entire microstructured region of each device were collected every hour for over 60 h. The start of the drying experiment (hour 1) was operationally defined as the time when the air interface had just reached the start of the microstructured region. The end of the experiment was operationally defined as three consecutive hours with no discernable change in moisture saturation. Image analysis followed a procedure similar to that described previously [19]. Generally, the microscopic images were stitched and exported from Nikon Elements software that operates the Nikon Ti-U microscope. The stitched images were first processed using open-source tools, including Python scripts, to mitigate image artifacts during the stitching process. In order to isolate the channel in the microscope images for analysis, the code from previous work [19] was modified (also available from https://github.com/pnnl/hydrogel_analysis/tree/master, Accessed on 7 November 2024). These modifications allowed importing a binary image mask that can isolate the individual microstructure regions from the rest of the no-geometry regions and inoculation ports. The saturation percentage was calculated using binary thresholding, specifically Otsu’s thresholding (available in the OpenCV Python package), to differentiate between pixels representing a “dry” area (air) and those representing “saturated” (liquid) or “solid” regions (soil particles). The number of “solid” pixels was determined by thresholding a completely dry micromodel (a stitched image of a dry device without liquid injected), and the saturation percentage was then calculated using the following formula:(1)$$p_{Sat}=\frac{A_{Sat}}{A_{T}-A_{So}}$$
where $p_{Sat}$ is saturation percentage, $A_{Sat}$ is the area of saturation, $A_{T}$ is the total area of the channel, and $A_{So}$ is the solid area (the micromodel design that represents soil particles).

### 2.5. Real-Time Mass Spectroscopy and Custom-Built Accessory Fabrication

The general assembly of the RTMS manifold for biological system monitoring has been reported elsewhere [9]. A custom mass flow controller was built to precisely regulate the flow within the microdevices. We specified a required flow range of 0.013 to 0.7 mL/min with an accuracy of (+/−)2% (Bronkhorst, Bethlehem, PA, USA). The schematic of the full assembly is available in Figure 1B. Custom acrylic accessories were fabricated using a laser etcher (Epilog, Golden, CO, USA) to connect the two-port microfluidics system with the flow controller. The top cover (Layer 1) is a narrow rectangle (1.5 × 10 cm) made of acrylic featuring two holes for screws and bolts (0.5 cm from each edge) and one section designed to accommodate both the inlet and outlet for capillary connections. This corresponds to the fungal ports from the ESMs and holds the capillary tubing perpendicularly, ensuring a sealed system. The inlet section was laser-etched with two circles (800 µm and 420 µm in diameter), and the outlet section was etched with one 420 µm circle. Layer 2 is a 0.5 cm thick cover made of 5:1 PDMS, designed to house the inlets and outlets. The inlet PDMS square was punctured with a 21-gauge, 0.81 mm (O.D.) blunt needle, and a 27-gauge, 0.42 mm (O.D.) blunt needle (Benecreat, Levallois-Perret, France), positioned 500 µm apart as the center. The outlet was punctured with a 0.42 mm (O.D.) blunt needle in the center. Layer 3 consists of a 0.5 cm cube made from a 5:1 PDMS mixture with a 4 mm punched hole serving as a connection bridge. A 1.5 cm square polyester track-etched (PETE) membrane (0.4 μm pore size, 12 μm thickness, 2 × 106 pores per cm^2^, 90 mm diameter, Sterlitech Corporation, Auburn, WA, USA) was trimmed and attached to the outlet port to prevent lid suction into the mass spectrometer and avoid disturbances, serving as Layer 4. The fungal microfluidics sample, mounted on a 2 × 3 glass slide, was positioned on the acrylic base as Layer 5. The bottom acrylic base (Layer 6) is a 10 × 13 cm rectangle with two drilled holes for bolts and screws. Two additional holes were drilled in the center of the acrylic base, aligned in a single column outside the microfluidics region.

To facilitate microfluid flow, a gas supply capillary (700 µm (O.D.), 530 µm (I.D.), Molex Polymicro, Lisle, IL, USA) was inserted through Layers 1 to 3 to introduce a pre-prepared air atmosphere with carbon dioxide removed into the well-controlled microsystem via the inlet port. A liquid supply capillary (360 µm (O.D.), 200 µm (I.D.), Molex Polymicro, IL, USA) was inserted through Layer 1 to the bottom of the fungal microfluidics platform, contacting the surface of the glass slide. The outlet capillary (360 µm (O.D.), 200 µm (I.D.), Molex Polymicro, IL, USA) was connected to the microflow controller, passing through Layers 1 to 3, bypassing Layer 4. Once all layers were stacked, bolts and screws were securely tightened from Layer 1 to Layer 6. A key requirement was ensuring that the PDMS layers were properly aligned and allowed clear visibility of the bottom geometry from a top view (see Figure 1C).

### 2.6. Rewetting of Fungal Microfluidics Using H_2_^18^O-Food Dye Mixtures

The liquid supply capillary was inserted into the micromodel during assembly for RTMS measurements. The capillary was connected to a 25 µL Hamilton syringe. A solution of blue food dye (Wilton, IL, USA) premixed with ^18^O-water (O^18^-Water > 98%, Cortecnet, Les Ulis, France) was used to enhance the visualization of liquid flow and test the hypothesis that fungi can utilize the introduced liquid source. A 7.5 µL mixture was injected once the system was connected to the RTMS and the system had been evacuated to establish a stable background. The liquid flow was driven primarily by the vacuum generated by the flow controller, which was connected to the RTMS system.

Fungal samples used in drying experiments under varying drought conditions were rewetted after the drought period, which was defined as the time when residual saturation remained stable for over 20 h. Typically, the drying process began on day 1 and ended on day 4. RTMS measurements taken between days 5–7 were classified as 0-week samples. Measurements conducted between days 12–14 were considered 1-week samples, and those scheduled for days 19–21 were designated as 2-week samples (see example images in Figure 2H–J)

## 3. Results

### 3.1. RTMS Micromodel Assembly Performance and Flow Testing

The successful operation of the apparatus relied on a multi-layer assembly on the opposing model ports (inlet and outlet) with secure capillary connections that enabled sustained atmospheric flow measurements over time. Connection to the RTMS through a purpose-built microflow controller provided a stable and static vacuum flow for atmosphere processing. To prevent leakage, the capillary was carefully inserted perpendicularly into the device, with laser-etched holes and an acrylic cover providing additional stability and alignment. The individually punched PDMS layers acted as effective barriers against leakage, while a higher mixing ratio of PDMS (5:1) cover enhanced model strength, ensuring secure capillary placement. PDMS bridges were incorporated to create a buffer zone around the outlet capillary, preventing liquid from reaching the mass spectrometer source, thus protecting the instrument from ionization filament damage. The attachment of a PETE membrane further minimized liquid from reaching the source, ensuring accurate data acquisition throughout the experiment. A three-channel micromodel was applied in this study to account for biological replicates. The placement of capillaries, the transport of fluids through the micromodel, and the speed of signal delivery to the instrument allowed consistency in this system, which was imperative for comparable measurement of the Birch effect.

### 3.2. Fungal Growth in the Microchannel

The fungal growth in micromodels was monitored using microscopic images (see example in Figure 2G). The devices were filled with a liquid medium, allowing the fungi to extend their hyphae from the fungal agar port into the pore spaces of the microchannels during incubation. A completely dark channel, resulting from epi-illumination, indicated full saturation during incubation. From day 0 to day 3, the hyphae from both fungal agars moved toward the empty channel, the areas before the soil-like structures, and started to branch and overlap once advancing to the soil region. By day 3 to day 6, the hyphae had infiltrated approximately halfway into the soil regions and potentially encountered hyphae extending from the opposite direction. Between day 6 and day 10, the fungi continued exploring the remaining pore space, eventually passing through the full extent of the channels. At this stage, the fungal hyphae had extensively explored within the pore spaces, and the additional incubation period (days 8–10) allowed further interaction between the fungi and the soil-like geometry. This extended period ensured sufficient fungal colonization and interaction with the geometry before the drying experiment. The incubation time of fungal growth was generally consistent across replicates, but some variability among replicates was observed. Factors such as the position of the incubation ports, as well as the placement and orientation of the fungal agar, contributed to inconsistent fungal biomass distribution on the surface across individual channels and devices. Fungal biomass was recorded for each replicate by comparing the weights of blank devices (six PDMS devices bonded to a 3 × 2” glass slide) with the weights of dried devices (six PDMS devices bonded to a 3 × 2” glass slide after drying), assuming the fungal biomass was distributed equally across each port and that remaining moisture and agar were negligible.

### 3.3. Drying and Rewetting of Fungal Biomass Within the Micromodel

To simulate drought conditions, the fungal microfluidics device was subjected to drying, during which moisture retention was closely monitored using time-lapse image acquisition. Liquid at the incubation port was removed first by dipping a small piece of Kimwipe tissue, and the device was left uncovered on the stage of the microscope without a PDMS square. Air infiltration occurred through the incubation port, the location with fungal agar inserted, following the path of fungal hyphae and spreading across the soil-like regions. The infiltration moved into the macro-pore channels and migrated the pore space of micro-aggregates of each channel, a pattern similar to what the previous studies reported [16,17]. The time required for saturation to drop below 50% was approximately 1 h for DI water at a controlled RH of approximately 80% [16]. With fungi incubated, the corresponding time to drop below 50% saturation under low relative humidity and temperature increased to 3–5 h at a controlled RH of 50%. The residual saturation was defined as the moisture that persisted after the evaporation of labile pore water. After 50 h of drying, the residual saturation stabilized at approximately < 20% from 30 to 50 h (see Appendix A). The illumination of microscopy across the stitched images might cause the variance of the residual saturation during image analysis. However, the residual moisture and fungal EPS accumulated within the soil micro-aggregates of the channel, preventing complete drying, as demonstrated previously [19], effectively replicating the natural water retention dynamics of soil pores.

Rehydration was initiated by injecting blue dye mixed with H_2_^18^O into the device, after assembly with real-time mass spectrometry (RTMS) accessories for CO_2_ efflux measurements. The liquid wicking through the capillary channels was dynamically influenced by sealing conditions, vacuum pressure, and the geometry of the tri-channels. Ideally, the liquid progressed uniformly through the tri-channels upon contact with the glass slide and fungal biomass within the microstructure. However, it occasionally wicked through inlet regions first, causing uneven distribution. At later stages, the liquid flowed synchronously through all channels, eventually reaching the end of the device at the outlet port.

### 3.4. Fungal Birch Effect from Soil Micromodel

Three types of fungal microfluidics models were prepared for this study, each subjected to different drying durations: 0-week dry, 1-week dry, and 2-week dry. The 0-week sample was measured for rewetting between days 5 and 7 after the start of the drying experiment; the 1-week sample was measured between days 12 and 14; and the 2-week sample was between days 19 and 21. All the measurements were conducted in biological triplicates. Despite the varying drought periods, the Birch effect was consistently observed across all fungal microfluidics models, characterized by a rapid surge in ^44^CO_2_ intensity following rewetting of ^18^H_2_O dye mixtures (see Table 1). The apex value, representing the peak of CO_2_ intensity and the initiation of the steady state, occurred when water wicked through the dried fungal biomass in the micromodel (reported in Table 2). The value was normalized by the fungal biomass measured before and after the drying experiment. The average apex value of ^44^CO_2_ was 2095 ± 2596 for 0-week samples, 2631 ± 2660 for 1-week samples, and 2157 ± 1969 for 2-week samples in Figure 3A. The residence time of apex for each treatment is also reported in Table 3 (0-week: 0.4 ± 0.19 min, 1-week: 1.4 ± 0.52 min, 2-week: 1.9 ± 1.5 min). Variations in the apex value and the residence time required to reach a steady state may have been influenced by flow conditions and the amount of ‘effectively’ dried, available fungi within the geometry as the liquid flowed across the surface of the glass slide. The outlet port connecting to the RTMS provided a consistent pulling force to drag the liquid once the blue water mixture entered; however, the liquid might not have fully covered the available fungal biomass on the surface during the initial movements. It later spread across the channels, traveling through the pore space and reaching the other side of the micromodel.

This initial ^44^CO_2_ release was followed by a second respiration event, triggered by the injection of H_2_^18^O mixtures. An example was plotted in Figure 3B. Generally, this second wave occurred within 10 min after the first burst of stored CO_2_ (see Table 4). In the 0-week sample, this ^46^CO_2_ wave occurred after approximately 10 ± 5 min, though one measurement was terminated early due to the detector saturation. The 1-week sample had this wave of ^46^CO_2_ at around 9 ± 0.6 min and the 2-week sample had the same phenomenon at around 8 ± 0.6 min across two replicates. This suggests that soil fungi, even under drought conditions, are activated to release stored CO_2_ within microbes upon rehydration. Additionally, the fungi absorbed the reintroduced water internally, leading to a subsequent wave of ^46^CO_2_ efflux triggered by such heavy water mixtures. This two-phase response illustrates the dynamic capacity of fungi to adapt to moisture availability, even after prolonged periods of drying.

## 4. Discussion

The primary goal of this study is to integrate real-time mass spectrometry (RTMS) with microfluidics to investigate fungal CO_2_ release upon rewetting within a soil-like platform with reduced complexity. Using an emulated soil micromodel that mimics the classical soil pore structure of the rhizosphere allowed us to visualize fungal growth at the pore scale, capture the drying behavior of fungal biofilms over time, and quantify the CO_2_ efflux from the fungal biomass upon rewetting. Establishing this microflow system demonstrates the first observation of measurable CO_2_ efflux in real time from axenic cultures of a soil fungus upon rewetting, confirming that microbial respiration at the pore scale plays a significant role in carbon release in drought-affected ecosystems.

### 4.1. RTMS Micromodel Assembly and Current Limitation

The efficiency of CO_2_ detection through the RTMS microfluidics system depends on several factors, such as the geometry of the microfluidics, vacuum stability inside the channels as well as active biomass being wet. Specifically, the three-channel device created fluid flow inconsistency upon rewetting, which resulted in unequal parts of the rewetting liquid reaching the outlet ports at variant times. Such inconsistency was driven by liquid wicking, capillary forces, selective vacuum channel flow, and surface hydrophobicity during the rewetting process. Eventually, all the fluids spread across the device reaching the outlet; however, unequal amounts of fluid flow were created during the measurement, leading to the variance in the apex value reported.

To ensure equal liquid delivery across the channels, the liquid capillary supply must be positioned and aligned in a centered location of the main port with an adequate liquid volume to cover tri-channels simultaneously. The delivery speed of liquid was further necessary to consider when initiating simultaneous flow through all three channels at the same time. One unknown parameter that requires further investigation is the effect of vacuum on the system itself. The flow through the system was established by the vacuum of the mass spectrometer, the pull force of the entire system, and regulated by the mass flow controller. To maintain flow through the system, a slight negative pressure was created in the environment surrounding the specimen. The system pressure depends on the flow through the connection between the capillary tubing and the micromodel. If there is a slight obstruction inside the system—such as a small PDMS residue or solid debris pulled by the vacuum and stuck in the capillary—the mass flow controller will proportionally increase the vacuum, based on the real-time response from the system and update dynamically, until the desired flow is achieved. Due to the micronized spec and delicate precision of the current system, it is challenging to effectively monitor pressure and the vacuum power applied to the entire system or the biomass.

### 4.2. Fungal Birch Effect and Its Drought Sensitivity

The well-defined and soil-mimicking microfluidics used in this study allowed us to inoculate axenic fungal cultures, visualize their growth and drying behavior, and quantify their contribution to CO_2_ release upon rewetting. Our findings showed a significant increase in CO_2_ release after rewetting, aligning with the Birch effect observed in prior studies [10]. Rather than attributing the CO_2_ release solely to bacteria or microbial communities from soil samples, this study is pioneering in demonstrating that soil fungi also contribute to CO_2_ efflux during drying and rewetting cycles. Moreover, soil fungi were able to internally utilize the introduced liquid from the local substrate and generate the following wave of CO_2_ release. The pronounced microbial CO_2_ efflux could be explained by the accumulation of osmolytes during dry periods, as hypothesized by Warren et al. (2016) [5]. Based on published studies, we hypothesized that the intensity of the drought period might contribute to the intensity of CO_2_ release. However, our normalized data and the reported table did not reveal a direct relationship between the amount of CO_2_ released and the drought period. There was no significant difference in the CO_2_ efflux between 0-, 1- and 2-week drought periods of the fungal cultures, as suggested by the *p* values. This suggests that the drought periods from 0 to 2 weeks do not have a dominant effect on CO_2_ efflux from axenic fungal cultures. The greater error bars may instead be attributed to technical limitations and the effective fungal biomass within the rehydration process as discussed above. A similar observation of variable CO_2_ release was reported in our previous study using soil collected from various geographical locations, showing that the intensity of CO_2_ was influenced by the microbial metabolism of the local microbial communities. A detailed analysis of respiration was not possible due to the granularity of the samples [9]. While the scale of the sample and the involvement of microbes in the studies were incomparable, such integration of RTMS with microfluidics, as presented in this study, may serve as an alternative method to uncover microbial contributions to the CO_2_ cycle as the next step.

### 4.3. Future Direction

To fully investigate the mechanisms of how fungi store carbon sources, such as the potential accumulation of osmolytes, and their conversion to CO_2_ upon hydration, labeled chemical monitoring is required as our next step. This could involve the ratio application of a ^13^C carbon source followed by the addition of H_2_^18^O to trace CO_2_ production, potentially complemented by an omics analysis of fungal biomass. Such an omics analysis would involve the extraction of fungal biomass at an endpoint and the extraction of metabolites and proteins to evaluate protein expression during drying and rewetting events. The expressed proteins and exuded metabolites would confirm biomolecular pathways that regulate biological cell viability upon drying and rewetting. The use of microfluidics offers a platform where soil properties can be precisely defined by computed porosities, particle distribution, and network complexity, and it also provides the possibility to modulate surface chemical properties, such as mineralogy. With the integration of RTMS, the controlled fluid composition can also be precisely managed and monitored during the process. Additionally, this step-by-step approach using microfluidics should shift to uncover microbial interactions in soil, focusing on how bacteria and fungi, the two dominant organisms in the underground ecosystem, work together to influence CO_2_ generation. These perspectives will provide pilot implications into the interactions between biotic and abiotic factors in CO_2_ flux for global carbon models where drying and rewetting cycles are common.

## 5. Conclusions

Overall, this study enhances our understanding of fungal involvement in soil carbon release, paving the way for a bottom-up approach to correlate pore-scale biogeochemistry to the macro-level CO_2_ efflux of the ecosystem. It represents the first successful integration of RTMS with a microfluidics platform to mimic soil systems, providing a simplified yet controlled environment for investigating pore-scale biogeochemical processes. Our findings demonstrate a sharp release of ^44^CO_2_ from soil fungi upon rewetting, followed by rapid internalization of water, regardless of prior drought duration, leading to a subsequent ^46^CO_2_ release. This innovative approach offers potential for future applications in monitoring gas composition during microbial activity, including volatile organic compounds, within a well-defined microenvironment, enhancing our understanding of pore-scale dynamics in biogeochemistry.

## 6. Patents

The RTMS-micromodel connection has been submitted for provisional patent. The application number is 63/707,706 titled as “MODELING LIVING SYSTEM METABOLIC RESPONSES IN MICROMODEL PLATFORMS.”

## Figures and Tables

**Figure 1 microorganisms-12-02295-f001:**
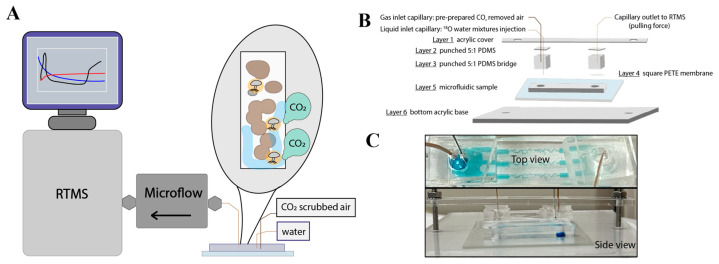
The overall scheme of this work and detailed assembly information. (**A**) Schematic representation of the experimental approach: integration of real-time mass spectrometry (RTMS) with a microfluidics model to investigate soil microbes at the pore scale. (**B**) Detailed schematic of the micromodel setup connected to the real-time mass spectrometry (RTMS) system. The diagram illustrates the interface between the microfluidics model and the RTMS. (**C**) Images of the real device assembly from both a top view (3 channels for liquid flow) and a side view, showcasing the setup and structural components of the microfluidics system integrated with the RTMS.

**Figure 2 microorganisms-12-02295-f002:**
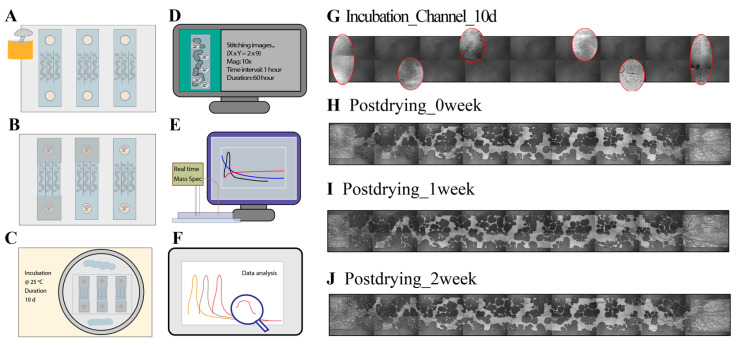
Workflow of this study and example images from monitoring. (**A**) Fungal inoculation into microfluidics. (**B**) Microfluidics covered with a PDMS square. (**C**) Incubation at 25 °C for 10 days. (**D**) Monitoring of drying experiments from microfluidics. (**E**) RTMS monitoring during rewetting of microfluidics. (**F**) RTMS data analysis. (**G**) Example images of fungi inoculated within microfluidics. The completely dark channel indicates full saturation after 10 days of incubation. The circled area highlights fungal movement within the geometry. (**H**) Example channel image of a 0-week dry sample. (**I**) Example channel image of a 1-week dry sample. (**J**) Example channel image of a 2-week dry sample.

**Figure 3 microorganisms-12-02295-f003:**
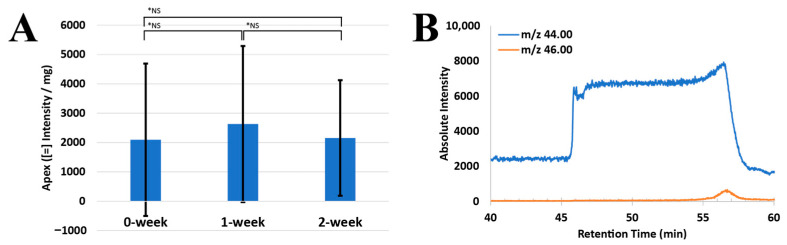
RTMS monitoring in a fungal microfluidics rewetting system. (**A**) Figure shows the apex (intensity per mg) of ^44^CO_2_ reported (*n* = 3). “*NS” indicates no significant difference (*p* > 0.05) between treatments. (**B**) Example monitoring of ^44^CO_2_ and ^46^CO_2_ upon hydration. Hydration was initiated at the 46th minute. During the first 5 min prior to hydration, both ^44^CO_2_ and ^46^CO_2_ showed stable signals. Upon rewetting the fungal microfluidics with a food dye and ^18^H_2_O mixture, the ^44^CO_2_ peak sharply increased. After approximately 10 min, the ^46^CO_2_ peak began to slowly rise, indicating fungal uptake of the reintroduced water. The experiment was terminated when the microflow was affected by moisture.

**Table 1 microorganisms-12-02295-t001:** Overall record of CO_2_ release monitored in this study.

Occurrence of Respiration After H_2_^18^O Injection			
^44^CO_2_\^46^CO_2_	0-Week	1-Week	2-Week
Rep1	+\− ^1^	+\+	+\−
Rep2	+\+	+\+	+\+
Rep3	+\+	+\+	+\+

^1^ Early termination due to detector saturation. "+" indicates observed respiration, while "−" indicates no respiration detected.

**Table 2 microorganisms-12-02295-t002:** The apex value of ^44^CO_2_ measured among various fungal samples upon rewetting.

Apex Value When Entering Steady State (Abs Intensity/mg)			
^44^CO_2_	0-Week	1-Week	2-Week
Rep1	141	364	322
Rep2	1105	5559	4238
Rep3	5041	1969	1912
Average	2095 ± 2596	2631 ± 2660	2157 ± 1969

**Table 3 microorganisms-12-02295-t003:** Residential time to enter a steady state after hydration of fungal samples upon rewetting.

Residential Time to Enter a Steady State After Hydration (min)			
^44^CO_2_	0-Week	1-Week	2-Week
Rep1	0.334	0.767	3.6
Rep2	0.6	1.65	1.117
Rep3	0.234	1.684	0.9
Average	0.4 ± 0.19	1.4 ± 0.52	1.9 ± 1.5

**Table 4 microorganisms-12-02295-t004:** Residential time to observe the respiration of ^46^CO_2_ upon rewetting.

Post-Hydration Time to Have the Respiration of ^46^CO_2_ (min)			
^46^CO_2_	0-Week	1-Week	2-Week
Rep1	N/A ^1^	8.48	N/A ^2^
Rep2	13.13	8.72	7.57
Rep3	6.13	9.53	8.41
Average	10 ± 5.0	9 ± 0.6	8 ± 0.6

^1^ Early termination due to detector saturation. ^2^ No observation of ^46^CO_2_ in this measurement.

## Data Availability

Dataset available on request from the authors.

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
