# Peer review of "Accessing Fungal Contributions to the Birch Effect: Real-Time Respiration from Pore-Scale Microfluidics"

_microorganisms, 2024, doi:10.3390/microorganisms12112295_

Round 1
Reviewer 1 Report
Comments and Suggestions for Authors
During drying/rewatering cycles, carbon in soil is released to the atmosphere as CO2; this phenomenon is known as Birch´s effect; microbial community in soils could have an essential role in this process. The manuscript describes the development of a technological approach based on Rela Time Mass Spectrometry to determine the release of CO2 in soil systems related to drying and rewatering. According to the result, the fungal strain (Fusarium sp. DS682) can use water supplemented during the rewatering process for metabolism. Supplemented water (H218O), employed in fungal metabolisms conducted to the release of 46CO2.
The research described in the manuscript is interesting, and the Real-Time Mass Spectrometry approach developed is novel and could be useful in determining key aspects of soil microbial metabolism. The authors need to address the following commentaries.
Main commentaries
No statistical analyses were conducted in the study, results are quite similar, and the error is great, I consider it important conduct statistical analyses and discuss the results. Explain how not identify significant differences among the treatments affect the description of the main findings and the discussion of results. The manuscript is strongest oriented to describe the developed device and their function, but it does not go into sufficient depth as to how it helps to understand the contribution of soil microorganisms to the Birch´s effect.
Additional commentaries
Lines 27-30, Complement information related to the contribution of the Birch´s effect on the release of CO2 (stored in soil) to the atmosphere (amounts, percentages respect all sources).
Line 41, add period in “[5,6] While”
Line 47, review if “from” is correct in “from rewetting”
Lines 52, 57, 93, 95, 96, 98,157, 158, 194, 252, 346, 465, and 474 review if “RTMS” is the adequate form, it is also used “RT-MS” choose adequate and be consistent in all manuscript
Line 63, eliminate extra space in “[10]. The”
Line 68, define acronym “EPS”
Line 69, review the fragment “availability to both local microbes” it is confusing
Lines 82, 102 correct format in fugal species “Fusarium sp.”
Line 107, 124, 127, 130, 138, correct centigrade symbol “oC”
Line 122, correct “cm^2” could be “cm2”
Line 128, eliminate extra space in “process. The”
Line 144, “1-cm square” could be “1 cm2”
Line 145, review “10-d” is presented in different forms such as “in line 159, 10 days” or “line 165, 10-day”
Line 188, eliminate comma
Line 197, add a space in “2%(Bronkhorst…”
Line 277, eliminate “direction. Between”
Author Response
Reviewer 1
During drying/rewatering cycles, carbon in soil is released to the atmosphere as CO2; this phenomenon is known as Birch´s effect; microbial community in soils could have an essential role in this process. The manuscript describes the development of a technological approach based on Rela Time Mass Spectrometry to determine the release of CO2 in soil systems related to drying and rewatering. According to the result, the fungal strain (Fusarium sp. DS682) can use water supplemented during the rewatering process for metabolism. Supplemented water (H218O), employed in fungal metabolisms conducted to the release of 46CO2.
The research described in the manuscript is interesting, and the Real-Time Mass Spectrometry approach developed is novel and could be useful in determining key aspects of soil microbial metabolism. The authors need to address the following commentaries.
Main commentaries
No statistical analyses were conducted in the study, results are quite similar, and the error is great, I consider it important conduct statistical analyses and discuss the results. Explain how not identify significant differences among the treatments affect the description of the main findings and the discussion of results. The manuscript is strongest oriented to describe the developed device and their function, but it does not go into sufficient depth as to how it helps to understand the contribution of soil microorganisms to the Birch´s effect.
Response-
We thank the reviewer for suggesting the statistical analysis. Upon performing the t-test on the datasets from week 0, 1 and 2 of drying, we find no significant differences between these conditions. This analysis between the datasets suggests that there are no significant changes to the Birch effect observed when an axenic fungal culture is dried and rewetted. Soil fungi significantly contribute to the transport of resources to host plants as well as bacteria during drought. As such, contributions of axenic cultures to the birch effect is essential to deconvolute contributions of abiotic and biotic soil components. The micromodel is a reduced complexity environment where contributions of biotic and abiotic soil components can now be investigated through the coupling of the micromodel platform with RTMS. The detailed statistical analysis is provided below.
Data analysis from Excel
Anova: Single Factor |
||||||
SUMMARY |
||||||
Groups |
Count |
Sum |
Average |
Variance |
||
0-week |
3 |
6285.89 |
2095.297 |
6738792 |
||
1-week |
3 |
7892 |
2630.667 |
7075358 |
||
2-week |
3 |
6472 |
2157.333 |
3878905 |
||
ANOVA |
||||||
Source of Variation |
SS |
df |
MS |
F |
P-value |
F crit |
Between Groups |
514514 |
2 |
257257 |
0.04362 |
0.957618 |
5.143253 |
Within Groups |
35386111 |
6 |
5897685 |
|||
Total |
35900625 |
8 |
|
|
|
|
t-Test: Two-Sample Assuming Unequal Variances |
||||||
|
0-week |
1-week |
||||
Mean |
2095.297 |
2630.667 |
||||
Variance |
6738792 |
7075358 |
||||
Observations |
3 |
3 |
||||
Hypothesized Mean Difference |
0 |
|||||
df |
4 |
|||||
t Stat |
-0.24949 |
|||||
P(T<=t) one-tail |
0.407635 |
|||||
t Critical one-tail |
2.131847 |
|||||
P(T<=t) two-tail |
0.81527 |
|||||
t Critical two-tail |
2.776445 |
|
||||
t-Test: Two-Sample Assuming Unequal Variances |
|
|||||
|
||||||
|
1-week |
2-week |
|
|||
Mean |
2630.667 |
2157.333 |
|
|||
Variance |
7075358 |
3878905 |
|
|||
Observations |
3 |
3 |
|
|||
Hypothesized Mean Difference |
0 |
|
||||
df |
4 |
|
||||
t Stat |
0.247706 |
|
||||
P(T<=t) one-tail |
0.408279 |
|
||||
t Critical one-tail |
2.131847 |
|
||||
P(T<=t) two-tail |
0.816558 |
|
||||
t Critical two-tail |
2.776445 |
|
|
|||
|
||||||
t-Test: Two-Sample Assuming Unequal Variances |
|
|||||
|
||||||
|
0-week |
2-week |
|
|||
Mean |
2095.297 |
2157.333 |
|
|||
Variance |
6738792 |
3878905 |
|
|||
Observations |
3 |
3 |
|
|||
Hypothesized Mean Difference |
0 |
|
||||
df |
4 |
|
||||
t Stat |
-0.03298 |
|
||||
P(T<=t) one-tail |
0.487637 |
|
||||
t Critical one-tail |
2.131847 |
|
||||
P(T<=t) two-tail |
0.975274 |
|
||||
t Critical two-tail |
2.776445 |
|
|
Additional commentaries
Response 2- We thank the reviewer for going through our manuscript in such great details. We have addressed these suggestions mentioned below.
Lines 27-30, Complement information related to the contribution of the Birch´s effect on the release of CO2 (stored in soil) to the atmosphere (amounts, percentages respect all sources).
Line 41, add period in “[5,6] While”
Line 47, review if “from” is correct in “from rewetting”
Lines 52, 57, 93, 95, 96, 98,157, 158, 194, 252, 346, 465, and 474 review if “RTMS” is the adequate form, it is also used “RT-MS” choose adequate and be consistent in all manuscript
Line 63, eliminate extra space in “[10]. The”
Line 68, define acronym “EPS”
Line 69, review the fragment “availability to both local microbes” it is confusing
Lines 82, 102 correct format in fugal species “Fusarium sp.”
Line 107, 124, 127, 130, 138, correct centigrade symbol “oC”
Line 122, correct “cm^2” could be “cm2”
Line 128, eliminate extra space in “process. The”
Line 144, “1-cm square” could be “1 cm2”
Line 145, review “10-d” is presented in different forms such as “in line 159, 10 days” or “line 165, 10-day”
Line 188, eliminate comma
Line 197, add a space in “2%(Bronkhorst…”
Line 277, eliminate “direction. Between”

Reviewer 2 Report
Comments and Suggestions for Authors
I am sending the review in the attachment.

Author Response
We thanks the reviewer for taking time to review our manuscript and providing encouraging comments

Round 2
Reviewer 1 Report
Comments and Suggestions for Authors
After reviewing the new version of the manuscript, I consider that the authors have adequately addressed the comments made in the previous version.
I consider no more crucial issues to address
Just check, in line 105, do not use italics for "sp" and in line 110, add a space between "°Cfor"